# Improved and optimized drug repurposing for the SARS-CoV-2 pandemic

**Sarel Cohen[1], Moshik Hershcovitch[2], Martin Taraz[3], Otto Kißig[3], Davis Issac◯[3]\*,
Andrew Wood[4], Daniel Waddington[5], Peter Chin[4], Tobias Friedrich[3]**

**1** The Academic College of Tel Aviv-Yaffo, Tel Aviv-Yaffo, Israel, **2** IBM Research, Haif, Israel, **3** Hasso Plattner Institute, University Potsdam, Potsdam, Germany, **4** Boston University, Boston, MA, United States of America, **5** IBM Research, Almaden, CA, United States of America

\* davisissac22@gmail.com

**Data Availability Statement:** Data are available at https://doi.org/10.5281/zenodo.7104738.

**Funding:** The author(s) received no specific funding for this work.

## Abstract

The active global SARS-CoV-2 pandemic caused more than 426 million cases and 5.8 million deaths worldwide. The development of completely new drugs for such a novel disease is a challenging, time intensive process. Despite researchers around the world working on this task, no effective treatments have been developed yet. This emphasizes the importance of *drug repurposing*, where treatments are found among existing drugs that are meant for different diseases. A common approach to this is based on *knowledge graphs*, that condense relationships between entities like drugs, diseases and genes. Graph neural networks (GNNs) can then be used for the task at hand by predicting links in such knowledge graphs. Expanding on state-of-the-art GNN research, Doshi *et al*. recently developed the D\textsubscript{R}-COVID model. We further extend their work using additional output interpretation strategies. The best aggregation strategy derives a top-100 ranking of 8,070 candidate drugs, 32 of which are currently being tested in COVID-19-related clinical trials. Moreover, we present an alternative application for the model, the generation of additional candidates based on a given pre-selection of drug candidates using collaborative filtering. In addition, we improved the implementation of the D\textsubscript{R}-COVID model by significantly shortening the inference and pre-processing time by exploiting data-parallelism. As drug repurposing is a task that requires high computation and memory resources, we further accelerate the post-processing phase using a new emerging hardware—we propose a new approach to leverage the use of high-capacity Non-Volatile Memory for aggregate drug ranking.

## 1 Introduction

With the novel coronavirus, a global pandemic with serious socio-economic implications for most parts of our daily lives is active [1]. The limited ability to take precautions for an unsuspected event like this and the rapid spread make finding an effective treatment as necessary as difficult, since the disease-specific knowledge is limited at the beginning and human lives are lost every day. Known and approved drugs happen to be well-studied, thus, they pose a good starting point for swift development of treatments, and an emerging tactic in fighting the

**Competing interests:** The authors have declared that no competing interests exist.

pandemic [2]. DrugBank, an extensive database compiling information about drugs approved by the US Food and Drug Administration as well as experimental drugs, contained more than 2300 approved drugs and over 4500 experimental drugs as of 2018; both with a strong upward trend [3]. This emphasizes the need for computer aided development of treatments.

Drug repurposing with knowledge graphs, as first described by [4], is the current state-of-the-art approach for finding possible treatments for novel diseases among known drugs using machine learning. Applying drug repurposing allows for a better way to maneuver through the pandemic. It can lead to better treatments for patients infected with one of the COVID-19 strains and a better understanding of the characteristics of the individual strains. Today, we approach the problem of drug repurposing using machine learning, focusing on deep learning methods. The idea of predicting unknown links between entities in a knowledge graph is traditionally known as *Collaborative Filtering*, as described by [5]. In this work we expand on the concept of *graph embeddings*, which map a fixed-size feature vectors to graph nodes and relations. A state-of-the-art technique for the creation of such embeddings based on deep neural networks (DNNs) is TransE [6].

Knowledge graph embeddings are already utilized to solve different tasks related to drug discovery, e.g., they are used to predict potential drug targets for diseases to reduce cost and increase speed in the drug development process in general [7]. Regarding the specific application of drug repurposing relying on edge prediction in a knowledge graph of biomedical data (see Section 2), [8] present a novel classification approach to this problem by implementing and merging various different ideas and techniques into one ensemble classifier. At its core, they deploy a DNN with an encoder-decoder structure. The encoder mechanism of it, which is based on the *Decagon* graph neural network by [9], was initially proposed for the prediction of side effects of concurrent drug use.

## Our Contribution

In this paper we extend the work done by Doshi and Chepuri [10]. Specifically we continue our work in Drug Repurposing [11, 12]. We offer the following contributions to the complex networks community analyzing medicine networks:

1. We introduce novel aggregation strategies to improve the post-prediction step of [10]. With these new aggragation strategies, we are able to obtain 50% more on the number of predicted drugs in the top-100 that were or are in clinical trials.

2. We re-implement the model described by [10] and improve it by allowing flexible neighborhood capture sizes. We also improve the implementation by [12] by improving training speed, inference time, readability and by reducing pre-processing time from 30 minutes to 2 minutes by leveraging matrix operations. We further extend the implementation to support Self-Label-Enhancement.

3. We explore the additional application of finding drug candidates similar to a manually preselected candidate using collaborative filtering on the same model output. We show that many drugs that are in clinical trial can be found by detecting the drugs that are the most similar (*e.g.* using cosine-distance on the embedding of the drugs) to a given known drug (or a subset of drugs) which is or was in clinical trials.

We also contribute to the *way* drug repurposing is computed. Drug repurposing is a task that requires large computational and memory resources. The emerging hardware of Intel Optane Persistent Memory Modules (Optane-PM) communicates via the memory bus, mitigating bottlenecks such as PCI-express lane availability, using the same interface as DRAM.

While there are other Persistent Memory technologies, Optane-PM being the most mature product on the market is based on 3D-XPoint (3DXP) technology and operates at a cache-line granularity with a latency of around 300ns [13], which is more than an order of magnitude faster than the current state of the art NVMe SSDs, but approximately three times slower than DRAM. Additionally, it has high capacity which is 8x larger than the available DRAM—a single DIMM of Optane-PM can reach 512GB. We note that it is practically necessary to use Optane-PM as the scale of the problem increases [14, 15].

To the best of our knowledge, in this paper, we show for the first time an application of the emerging Optane-PM for the task of Drug Repurposing. We generate a large dataset for the Drug Repurposing problem by extending (both vertically and horizontally) the dataset we have and evaluate two simple aggregation strategies which are implemented and processed on the Optane-PM. We obtain fast and promising results for the use of Optane-PM to process large datasets in the context of Drug Repurposing.

## 2 Dataset

Our work relies on the Drug Repurposing Knowledge Graph (DRKG) by [16], which compiles data from different biomedical databases. It contains 97, 238 entities belonging to 13 entity types and 5, 874, 261 triplets belonging to 107 edge types. We restrict ourselves to 98 edge types between 4 entity types, namely gene, compound, anatomy and disease, which leaves us with a knowledge graph with 69, 036 entities and 4, 885, 854 edges. In particular, it contains drugs and substances as *compound* entities, as well as different COVID-19 variants as *disease* entities. There are 8,070 drug entities and 33 different COVID-19 entities. The edge types include e.g. *compound-treats-disease* edges, which is the kind of edge our model predicts.

One part of DRKG are the precomputed TRANSE embeddings trained using dgl-ke by [17]. To train our model to predict whether a given edge in some *compound-treats-disease* relation exists, we have to create suitable training data. To provide our model with both positive and negative samples for training, for each positive edge we sample 30 non-edges in the dataset, which results in a ratio similar to DR-COVID. This process tries to account for the imbalance of edges and non-edges in the ground truth. The set of edges included in the dataset is not complete, however, it is quite certain to be correct. Consequently, the positive edges are given a higher weight in the loss calculation, and the higher number of negative edges (which are not certain to be truly negative) are given a lower weight. To prevent too much imbalance in the individual minibatches, we use a weighted random batch sampler that over-samples the positive samples yielding an expected ratio of 1 : 1.5 of positive to negative samples in each batch.

## 3 Model architecture

A Graph Neural Network (GNN) is a message passing framework where vertex embeddings are passed along edges of a graph. A single GNN layer traditionally performs a single round of message passing where messages are transformed via an *edge function*, are collected together into a single message via an *aggregator function*, and finally are used to produce new messages using a *vertex function*. We refer the reader to [9, 18, 19] for a more in-depth description.

In our experiments, we used a traditional encoder-decoder architecture using a two-layer GNN encoder and a custom decoder. The architecture of our model is illustrated in Fig 1. It consists of a SIGN [20] architecture encoder, which provides an embedding $y \in \mathbb{R}^{250}$ for each node. We apply *tanh* to the encoder output and forward it into our decoder. Given two nodes $u$, $v$, the decoder takes their encodings $y_u, y_v$ and assigns a score $s_{u,v} \in [0, 1]$, which measures the probability for an edge between nodes $u$ and $v$ to exist. The decoder consists of two linear

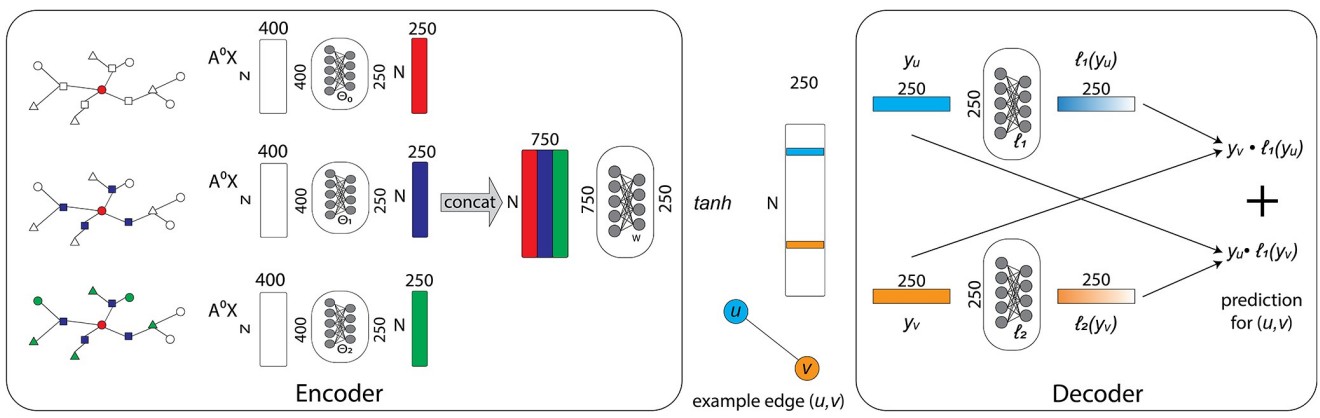

**Fig 1. Model architecture.** The architecture of our model as described in Section 3.

layers $\ell_1(u)$ and $\ell_2(v)$ that process the encodings $y_u$ and $y_v$ via a sigmoid function, that is, $\sigma(y_v \cdot \ell_1(y_u) + y_u \cdot \ell_2(y_v))$. The loss of the model is computed using a binary cross entropy loss with logits with weights set as described in Section 2.

## Implementation

The dataset presents itself as a list of triples, each posing source, relation-type and sink of an edge. This is accompanied by precomputed knowledge graph embeddings. For the preprocessing we first filter out the edges belonging to the part of the knowledge graph we restrict ourselves to. We then construct a graph with the help of DGL [21]. To compute the neighborhood embeddings we feed into the model, we first derive an adjacency matrix $A \in \{0, 1\}^{n \times n}$ from the reduced graph, from which the edges we try to predict, i.e., *compound-treats-disease* edges, have been removed. We then derive the normalized graph Laplacian $\tilde{A} = D^{-\frac{1}{2}} A D^{-\frac{1}{2}}$ where $D_{i,i}$ is the degree of node $i$. Suppose $X \in \mathbb{R}^{n \times 400}$ is the matrix of graph embeddings for the $n$ nodes, then the $k$th neighborhood is defined as $\tilde{A}^k X$.

## 4 Output interpretation

In this section we present different strategies for interpreting the scores that the model outputs for the application of predicting the top-$r$ most promising drug nodes for a given set of disease nodes $D$. Note that this is important as there are multiple COVID-19 diseases. Let $n$ be the total amount of drug nodes. Predicting all $n \cdot |D|$ edge combinations, our model yields a matrix of scores $S \in \mathbb{R}^{|D| \times n}$. Note that in our experiments, we have $|D| = 33$, $n = 8070$, and $r = 100$. For each of the following strategies we first perform a standardization of the scores per disease using $\hat{s}_{dc} = \frac{s_{dc} - \mu(s_{d*})}{\sigma(s_{d*})}$, where $d$ is the index of a disease in $D$, $c$ being the index of the drug, $\mu(s_{d*})$ and $\sigma(s_{d*})$ denote the mean and standard deviation over all drugs.

Certain "mild" diseases may be affected by plenty of drugs resulting in those being linked more likely. The standardization helps to achieve a better comparability across different diseases, allowing us to identify the suited drugs for every disease individually and compare those. However, this could also give good scores to some drugs in the case of diseases with no "good" scores in the first place, potentially yielding some less useful proposals.

An aggregation strategy takes our matrix of standardized scores $(\hat{s}_{dc})$ and derives a list of drugs from it, the top-$r$ of which are our result. We propose the following aggregation strategies. For **global score mean**, we calculate the means of $(\hat{s}_{dc})$ along axis 0, that is, over all

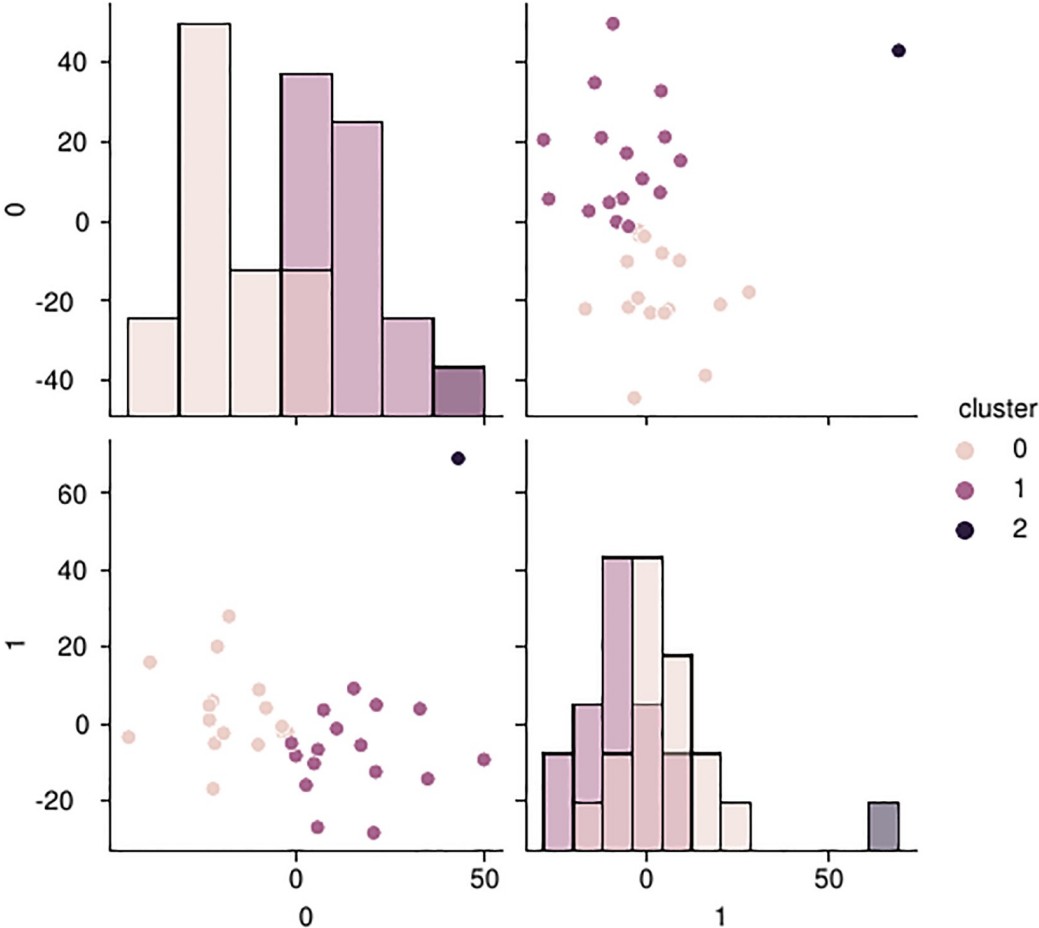

**Fig 2. PCA clustering.** Pair plot showing the population of COVID-19 diseases in our dataset by their standardized scores reduced from *n* dimensions (the number of drugs) to two with the help of a PCA. The colors show the affiliation to a cluster derived from the application of a 3-Means clustering.

diseases per drug; then we sort the drugs by their respective scores and select the top-*r*. For **global score maximum**, we find the maxima of $(\hat{s}_{dc})$ along axis 0; then again we sort the drugs and select the top-*r*. For **union over disease rankings**, we calculate top-*x* drugs per disease with *x* as small as possible such that we get at least *r* unique drugs in the union. We then concatenate all those top-*x* lists together to get a top-*r* drug list.

Furthermore, in **k-means score maximum**, grouping similar disease types can be used to enhance the accuracy of our top-*r* predictions. We perform such a grouping using the k-means clustering algorithm. For each cluster, which now represents a group of similar diseases, we use a mean reduction to calculate the score of a drug and then reduce to the maximum across these clusters. A sensible number of clusters to create can be chosen by performing a principal component analysis (PCA) [22] on the standardized scores. This allows us embedding data points from $\hat{s}_{dc}$ into the 2d plane as shown in the pair plot in Fig 2. This visually suggests that 3 clusters among the diseases exist, which can be picked up by a 3-Means clustering. Then, for **union over k-means rankings**, we perform the top-*x* selection on clusters calculated with the clustering method described above. This not only allows us to use a greater *x* because

we have fewer lists to pick from, but also to get more consistent top picks because of the internal averages that we apply inside each cluster.

We also do a network based clustering approach where we substitue k-means in the above strategy with the louvain clustering algorithm [23]. We call these **louvain score maximum** and **union over louvain rankings**. The network clustering is applied by viewing the prediction matrix as a weighted bipartite graph. Since louvain requires positive weights, we threshold the weights and give 0 weight to each entry that is negative. This sparsifies the network and helps to create more network structure. In louvain clustering, the number of clusters are inferred by the algorithm by assessing the modularity of the network. The purpose of using such a clustering method is to take advantage of any network structure that is present in the output, that k-means might not take into account.

For **drug clusters mean** strategy, we apply a k-means algorithm to the drugs (as opposed to diseases in above strategies) with the standardized score vectors over the diseases as datapoints. We set the parameter $k$ for $k$-means to be rather high so that the cluster sizes are on average much smaller than $r$. We then rank clusters according to their mean score. We pick the top clusters until we reach drug size of $k$. To break the ties within the last picked cluster, we use the mean score of the individual drugs in the cluster.

Another strategy we apply is the **mean with $\ell$ outliers** strategy where for each drug, we take the mean across the best $|D| - \ell$ of its score entries. This means we consider that $\ell$ of the diseases could be outliers for a drug and ignore them.

We also apply three strategies that are based on finding bilciques in the weighted bipartite graph formed from the prediction matrix. Here, the two bipartitions are naturally the drugs and diseases. We enumerate over all subsets of the disease-set and for each such subset $S$ of diseases, we find the value of the maximum biclique induced by $S$, calculated as follows: for each drug $d$, let $\mathtt{sum}(d, S)$ denote the sum of weights of edges of $d$ to $S$; let $d_1, d_2, \ldots$ be the order of drugs such that $\mathtt{sum}(d_i, S)$ is decreasing; let $i$ be the index for which $i \cdot \mathtt{sum}(d_i, S)$ is maximized, then the max biclique induced by $S$ is given by $(\{d_1, d_2, \ldots, d_i\}, S)$, and its value is $i \cdot \mathtt{sum}(d_i, S) \cdot |S|$. Then we pick the subset $S^*$ with the largest maximum biclique value, and rank the drugs on the decreasing order of their sum of weights of edges into $S^*$. Since we need to enumerate the subset of diseases, it is timewise expensive to do this on the whole disease set. So, we cluster the diseases into 3 clusters by using 3-Means and then aggregate the ranking over the clusters, either by picking the maximum or the top-x strategy. Depending on the aggregation strategy, thus we have two biclique-based strategies, i.e. **biclique maximum** and **union over biclique rankings**. We also do a third simpler biclique-based strategy, where we do not enumerate over diseases, but calculate the max biclique induced by the whole set $D$ of diseases. We call this much faster strategy as **cumulative max**.

## 5 Collaborative filtering

Suppose we already have pre-selected some candidates for clinical trials. Now we would like to identify similar candidates that could be interesting. This new application can be approached using collaborative filtering on our model output. We measure the similarity along the model's edge predictions per drug. To precisely define the cosine similarity between two given drugs $i$, $j$, let $\hat{s}_{*i}, \hat{s}_{*j}$ be their prediction scores along the disease dimension. Then their similarity is defined as $\hat{s}_{*i} \cdot \hat{s}_{*j}$. We test this application by ranking the remaining drugs of our dataset by the cosine similarity to pre-selected candidates. Our pre-selections are sampled randomly from the clinical trial dataset.

## 6 Experimental evaluation

### 6.1 Methodology

We first train our link prediction model to generate probability scores to a candidate edge using an encoder-decoder architecture described in Section 3. We implement the model using PyTorch. We train it using the Adam optimizer [24]. We use 90% of the data for training and the rest for validation. The training is performed on Google Colab utilizing a Nvidia Tesla T4 and it takes ∼ 2 minutes to prepare the graph dataset. We train our model using 25 epochs with a starting learning rate of $10^{-5}$ and a weight decay of $10^{-2}$. Each training epoch took us 30 seconds, which is a significant improvement over the 610 seconds of the implementation by [10] and can be attributed to the exploitation of data parallelism we added.

Using this model, we generate prediction matrices by sorting drugs for each covid strain using the learned model. We then use aggregation strategies to process the prediction matrix to determine the final drug rankings. These drug rankings are then compared against drugs which are being currently tested in clinical trials.

### 6.2 Link prediction and drug aggregation performance

To test our link prediction model, we compare the top-100 drugs for SARS-CoV2 computed by our learned model to those predicted utilizing the weights of [10]. While their model's top-100 predictions include 22 drugs present in clinical trials, we only reach 15. We suspect the hand-made adjustments to the dataset utilizing undisclosed data sources are responsible for this discrepancy, as this is the sole missing part in our implementation. Consequently, we use their published rankings to measure different aggregation strategies. Fig 3 plots the prediction matrix and highlights the scores of the drugs in clinical trials. It is easy to observe from the plot that there is a high correlation between being in clinical trial and having a high score in our predictions.

To test drug aggregation strategies, we use each strategy to combine rankings of each drug for each covid type to produce a final top-100 ranking. We then compute the number of intersections with the drugs that are currently the subject of clinical trials related to COVID-19 [25]. This information is available on Kaggle as a list of drug names [26] and contains 250 drugs.

The results of the the different aggregation strategies can be found in Table 1. We see that our Union over Cluster Rankings with KMeans(k = 3) outperforms the other approaches, yielding 32 hits. This is intuitive as using PCA on the prediction scores shows that there are three clusters among the COVID strains. In contrast, DR-COVID's aggregation method, Union over Disease Rankings, reaches just 21 hits in our evaluation process. In Table 1, we also provide ther running times of the aggregation strategies while running them on Google Colab.

We also compare with another previous drug repurposing work [27], that used an aggregation of different predicting algorithms to predict a list of top-100 drug candidates for Covid-19. Their list of top-100 drugs hits only 22 drugs in the clinical trial list compared to 32 of our best strategy.

In Table 3 in S1 Appendix, we give the list of drug names in clinical trials that are hit in the top-100 rankings of five of our aggregation strategies. We select the five strategies that are different in approach to each other and give top number of hits as per Table 1. The five strategies we select are Mean with 2 outliers, Union over k-means cluster Rankings (k = 3), Louvain cluster Score Maximum, Drug clusters mean with KMeans(k = 250), and Union over Biclique rankings. In Table 4 in S1 Appendix, we give the rankings of drug names in clinical trials for

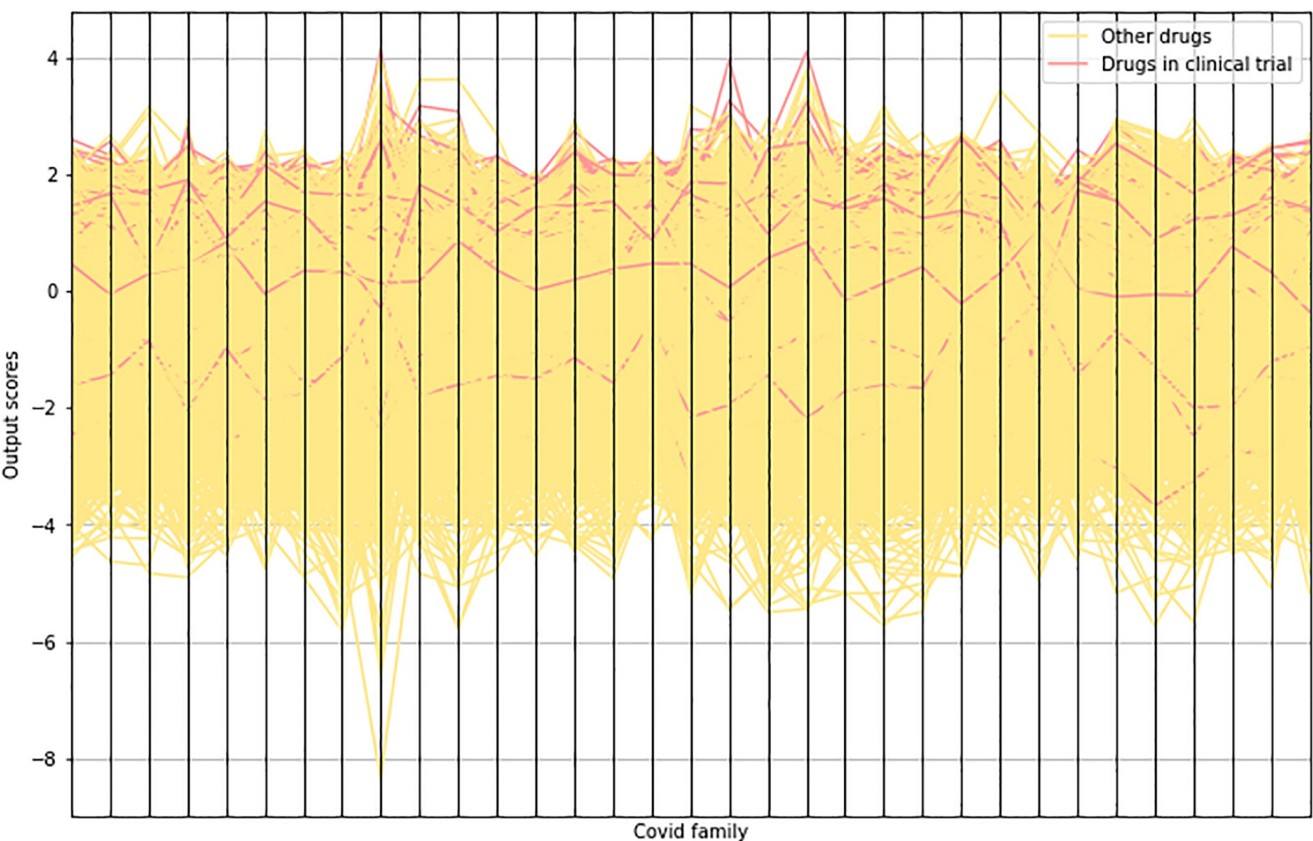

**Fig 3. Link prediction scores.** Parallel coordinates plot showing the standardized scores output by our link prediction model. We highlight the drugs that are present in clinical trial. The plot shows a high correlation between the scores and being in clinical trial.

**Table 1. Hits of proposed candidates in actual clinical trials.**

| Aggregation strategy | # hits | Time(s) |
|---|---|---|
| Single Disease (median) | 20 | 0.26 |
| Global Score Maximum | 22 | 0.852 |
| Global Score Mean | 30 | 1.228 |
| Mean with 2 outliers | 31 | 0.376 |
| Union over Disease Rankings (Dʀ-COVID, [10]) | 21 | 1.45 |
| K-Means cluster Score Maximum (k = 8) | 18 | 1.343 |
| K-Means cluster Score Maximum (k = 3) | 20 | 0.561 |
| Union over k-means cluster Rankings (k = 8) | 24 | 1.256 |
| Union over k-means cluster Rankings (k = 3) [12] | **32** | 1.227 |
| Union over Louvain Cluster Rankings | 27 | 1.153 |
| Louvain cluster Score Maximum | 29 | 1.24 |
| Drug clusters mean with KMeans(k = 250) | 30 | 20 |
| Drug clusters mean with KMeans(k = 500) | 28 | 27 |
| Biclique Max | 26 | 1560 |
| Union over Biclique rankings | 31 | 1500 |
| Cumulative Max | 26 | 0.035 |

each of these five strategies (for those in top-100 ranks). In Table 5 in S1 Appendix, we give the list of all drugs that are *not* known to us to be in clinical trial but are present in the top-100 rankings of all of the above five strategies. We observe that *Cidofovir*, the top entry in this table, i.e. the drug that is not in clinical trial but is best ranked by our strategies, was also independently found by another study to be promising for Covid-19, while unfortunately also having negative side-effects [27]. The study goes on to suggest alternatives with similar structure as Cidofovir instead.

We observe that hits are not evenly distributed along the rankings of the aggregation strategies, with more hits towards rank 60 and higher, suggesting we are unlikely to get better results by predicting more than the top-100 drugs.

## 6.3 Collaborative filtering performance

In the case of one single pre-selected candidate, for selecting the top-100 drugs ranked by similarity to the pre-selected candidate we get a mean of 18 (min. 0, max. 32) hits. Conducting the experiment with 15 pre-selected candidates and selecting drugs corresponding to the top-100 of a global ranking of all similarities yields on average 18 (min. 0, max. 37) hits.

## 7 Accelerating drug repurposing by using NVRAM

In this section we outline a new approach for using the emerging Non-Volatile Memory for analyzing large datasets for the task of drug repurposing. We believe such technologies can have a high impact on these big-data tasks. More specifically, we demonstrate the use of Non-Volatile Memory for aggregate drug prediction. In general, Optane-PM can perform arbitrary matrix calculations while providing significantly more capacity than ordinary DRAM and optionally providing persistence. We chose to use Optane-PM to implement the Global Score Mean and Global Score Maximum aggregation strategies. We chose these strategies for their decent prediction performance (see Table 1) and because they were easy to implement using Optane-PM. We note that we did not select strategies which used clustering due to an incompatibility between scikit-learn [28, 29] (the package clustering was implemented with) and the Optane-PM library. We show that by using Optane-PM, we can process datasets faster than with traditional storage methods such as DRAM + NVMe SSD or memory mapping.

To demonstrate the utility of Optane-PM, we artificially increased the size of the data being operated on. To do so, we extended our ranking matrix of size 2MB by concatenating entries both vertically and horizontally. Using this scheme, we created data matrices of sizes 33, 66, 131, and 261GB. This was necessary to show the performance difference between Optane-PM and other storage methods.

### 7.1 Interacting with Optane-PM

We use a Python 3 library called PyMM [30] to interface with Optane-PM. PyMM has been developed as part of the Memory Centric Active Storage (MCAS) system [31]. PyMM provides a set of abstractions and framework for managing Python variables in locally-attached Optane-PM. For more details regarding MCAS and PyMM, we direct the reader to [14, 15, 30, 31]. Data that is stored in PyMM is persistent and can be accessed and manipulated in-place, directly on device, without requiring a copy or transfer to DRAM. Using PyMM, we store our large data matrices and create aggregate rankings using the two strategies mentioned in Section 4.

## 7.2 Experiments

We compare Optane-PM against other storage methods. We measure the runtime of using different (simple) aggregation strategies implemented on Optane-PM against their implementations using DRAM. We augment the prediction matrices using the procedure mentioned in Section 7 to produce arbitrarily large data. Our experiment compares the following implementations:

1. **PyMM implementation**. Prediction matrices are stored on Optane-PM and are processed on device using the Global Score Mean and Global Score Maximum strategies.

2. **DRAM implementation**. Prediction matrices are stored entirely on NVMe SSD, and then transferred and processed in DRAM. We note that this implementation is only possible if the machine has sufficient DRAM.

3. **MMAP_384 implementation**. Prediction matrices are stored on NVMe SSD. During processing, the required data is loaded from NVMe SSD to DRAM using NumPy's Memory-Mapping functionality. In this implementation we have 384GB of DRAM. This configuration allows the entire dataset to be loaded into DRAM, meaning no evictions will occur. Therefore, this is a best-case scenario for memory mapping performance.

4. **MMAP_64 implementation**. This implementation is a more realistic memory mapping scenario. While this implementation is almost identical to the previous one, the amount of DRAM has been restricted to 64GB. This means that the memory mapping routine will need to evict data from DRAM during processing. From a Cloud/infrastructure perspective, this simulates a low-cost machine memory mapping scenario.

Our experiments were conducted on a Lenovo SR650 2U server equipped with two Intel Xeon Gold 6248 (2.5GHz) processors supporting 80 CPU hardware threads. The server is also equipped with 384GB (12x32GB) of DDR4 DRAM and 1.5TB of Optane-PM (12x128GB) as well as two NVMe SSD disks with 3TB each.

## 7.3 Scalability results

Our implementation results can be seen in Table 2 and Fig 4. Specifically we note that while DRAM operates with lower latency than Optane-PM, using DRAM involves expensive copy operations while Optane-PM(configured persistently) does not. This is why, in our experiments, Optane-PM always outperformed DRAM.

Specifically, our results include the cost of loading data from disk (as needed). The loading time is non-trivial (see Table 2). One advantage of Optane-PM is that data is persistent and has no loading time. To measure the cost of compute only, we also tracked the running time after data was loaded. In this case, using Optane-PM is between two and three times slower than DRAM, which is expected since the latency of Optane-PM is known to be approximately three times that of DRAM. We note that the slower latency of Optane-PM is well worth the trade-off for higher capacity as well as persistence.

**Table 2. The time (in seconds) to copy data from NVMe SSD to DRAM as a function of DRAM size.**

| DRAM load time (sec) | | | |
|---|---|---|---|
| **33GB** | **66GB** | **131GB** | **261GB** |
| 13.5 | 26.3 | 83.6 | 203.5 |

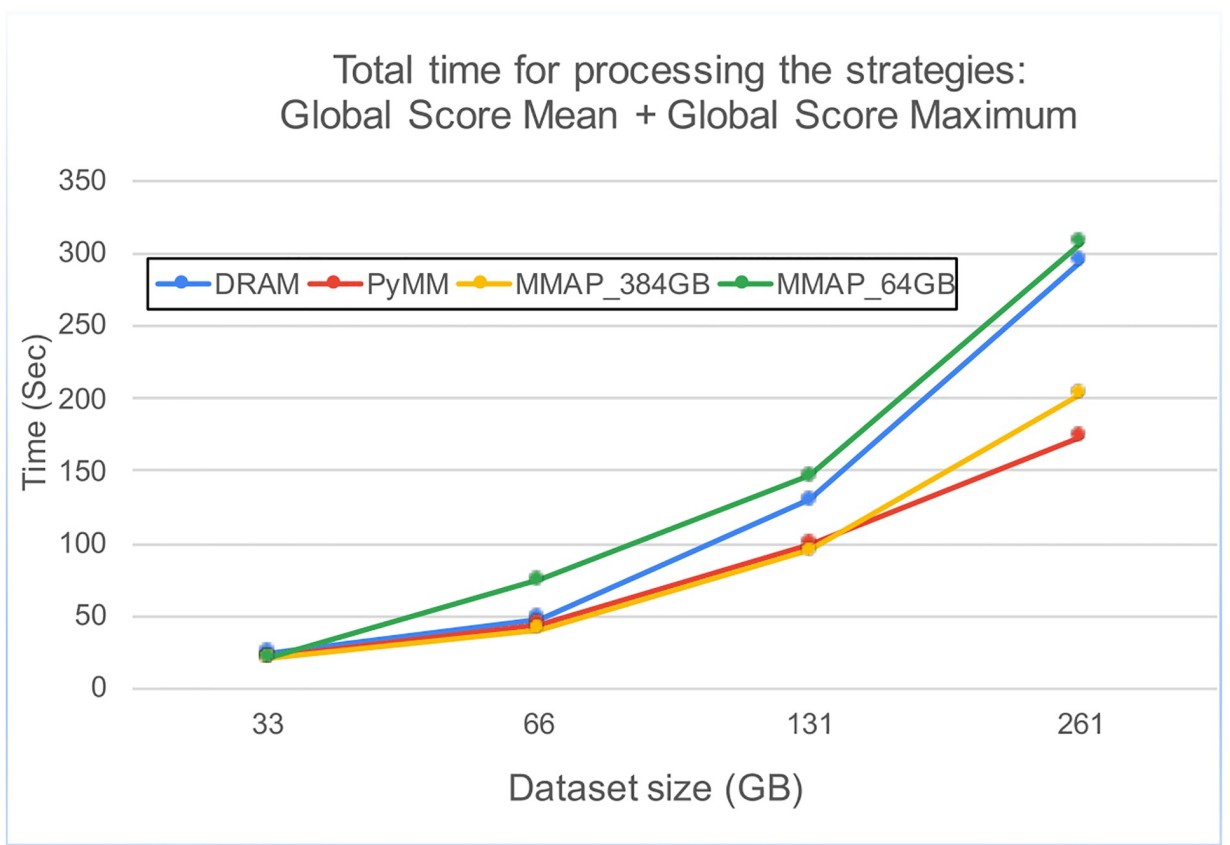

**Fig 4. Strategy processing times.** The total time for processing the two strategies: Global Score Mean and Global Score Maximum one after another.

Surprisingly, the best-case memory mapping implementation performed almost as well as the Optane-PM implementation. This is an artifact of our drug rank aggregation strategies. The two strategies we evaluate are single scan operations, which behaves efficiently using memory mapping. As the memory size increases, we observe performance degradation when memory exceeds 192GB. This is a result of the dual cpu architecture: DRAM is split between the two sockets in a Non-Uniform Memory Access (NUMA) architecture. This means that after 192GB (the maximum amount of DRAM allocated to a single socket), data must cross to the other socket which induces additional latency.

In a more realistic setting, memory mapping performs the worst. This is due to the eviction policy and DRAM not being able to store the entire dataset. This can already be seen in the memory-mapping strategy with 64GB in Fig 4. We note that this is also a best-case realistic scenario as once evicted, a row will never be needed again by our strategies. For more advanced strategies, memory mapping will perform significantly worse as multiple passes (sometimes random access) of the data is required.

In general, we observe that it is advantegous to use Optane-PM when working on tasks such as drug repurposing. Despite longer latencies than DRAM, our aggregation strategies benefit from the higher capacity and persistence offered by Optane-PM and avoid classical limitations of using DRAM such as expensive copy/load operations and NUMA boundaries. We also observe that Optane-PM is vastly superior to existing solutions such as memory mapping when data exeeds DRAM capacities.

## 8 Conclusion

Deep learning can help the development of drugs in the face of a global pandemic. Rather than looking for promising candidates by hand, one can instead rely on graph neural networks. We have been able to clarify the evaluation part of Dʀ-COVID [10] and proposed an aggregation technique yielding better results. Our own implementation improves both training speed as well as readability. We have also shown that using Optane-PM allows researchers to scale techniques efficiently to large datasets, which benefits the drug repurposing community.

## Supporting information

**S1 File.**
(DOCX)

**S1 Appendix.**
(PDF)

## Author Contributions

**Conceptualization:** Sarel Cohen, Moshik Hershcovitch, Andrew Wood, Daniel Waddington, Peter Chin, Tobias Friedrich.

**Data curation:** Davis Issac.

**Investigation:** Martin Taraz, Otto Kißig.

**Validation:** Davis Issac, Andrew Wood.

**Visualization:** Davis Issac.

**Writing – original draft:** Davis Issac, Andrew Wood.

**Writing – review & editing:** Davis Issac, Andrew Wood.

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
