## [Decision Letter · Decision Letter 0]

3 Jun 2022

PONE-D-22-06101Improved And Optimized Drug Repurposing For The SARS-CoV-2 PandemicPLOS ONE

Dear Dr. Isaac,

Thank you for submitting your manuscript to PLOS ONE. After careful consideration, we feel that it has merit but does not fully meet PLOS ONE’s publication criteria as it currently stands. Therefore, we invite you to submit a revised version of the manuscript that addresses the points raised during the review process.

We look forward to receiving your revised manuscript.

Kind regards,

Carlo Vittorio Cannistraci

Academic Editor

PLOS ONE

Journal Requirements:

2. Please amend the manuscript submission data (via Edit Submission) to include author Andrew Wood

3. Thank you for submitting the above manuscript to PLOS ONE. During our internal evaluation of the manuscript, we found significant text overlap between your submission and the following previously published works, some of which you are an author.

- https://link.springer.com/chapter/10.1007/978-3-030-93413-2_61

Please revise the manuscript to rephrase the duplicated text, cite your sources, and provide details as to how the current manuscript advances on previous work. Please note that further consideration is dependent on the submission of a manuscript that addresses these concerns about the overlap in text with published work.

Additional Editor Comments:

I read the article myself and I read also the Reviewer 1 comments.

As an editor I suggest the following:

1. If the Authors want to propose a novel approach of drug repurposing.

I agree with Reviewer 1 that, in this case authors should compare the approach with respect state of the art, while they compare only with a previous approach of the same group.

For instance, if applicable, the authors can consider to compare with other methods in literature that are of relevance and reported here:

Network medicine framework for identifying drug-repurposing opportunities for COVID-19

Deisy Morselli Gysi et al. PNAS 2021

Pioneering topological methods for network-based drug–target prediction by exploiting a brain-network self-organization theory

C Durán, S Daminelli, JM Thomas, VJ Haupt, M Schroeder, ...

Briefings in bioinformatics 19 (6), 1183-1202

I want to stress that to accept your study, I do not need that your proposed method performs better than the ones you compare.

I only require a fair comparison that shows and discusses the level of performance in respect with other methods that are considered state of art in literature.

This is important to evaluate your proposed method in the context of the current scientific literature.

2. If the Authors want to present an efficient and novel implementation of a previous approach.

I disagree with Reviewer 1 that in this case probably PLOS one is not the best option (since authors should present advancements in computer science and high performance calculus).

I believe that a new efficient implementation of a previous approach is of interest for our journal if the Authors can provide/discuss quantitative and relevant evidence at support of the new implementation

in comparison to the previous, but also compare with other methods not considered in previous study of the same group, such as the one I mentioned at the point above.

3. I agree with Reviewer 1 that the authors should clarify the context in the introduction and then to modify results and discussion consequently, since they did a hard work, and we are interested to publish their article.

Goodluck with your revision

best

Carlo Vittorio Cannistraci

Comments from Staff Editor: We note that the academic editor has recommended that you cite specific previously published works. As always, we recommend that you please review and evaluate the requested works to determine whether they are relevant and should be cited. It is not a requirement to cite these works. We appreciate your attention to this request.

Reviewers' comments:

Reviewer's Responses to Questions

**Comments to the Author**

1. Is the manuscript technically sound, and do the data support the conclusions?

Reviewer #1: Yes

2. Has the statistical analysis been performed appropriately and rigorously? 

Reviewer #1: Yes

3. Have the authors made all data underlying the findings in their manuscript fully available?

Reviewer #1: Yes

4. Is the manuscript presented in an intelligible fashion and written in standard English?

Reviewer #1: Yes

5. Review Comments to the Author

Reviewer #1: Authors present an interesting approach for drug repurposing. The approach is based on graph neural network, python programming language and high performance platforms. The approach is technically sound and it is nicely written. References are also adequate. Despite this, in my opinion paper has a main weakness due to lack of the clarity of the message. Do the authors present a novel approach of drug repurposing? . In this case authors should compare the approach with respect state of the art, while they compare ongly with a previous approach of the same group. So in this case paper should be rejected. Alternatively, the authors want to present an efficient and novel implementation of a previous approach. In this case probably PLOS one is not the best option, since authors should present advancements in computer science and high performace calculus. I know that this a common problem in all the multidisciplinary work, so i suggest to the authors to clarify the context in the introduction and then to modify results and discussion consequently, since they did a hard work.

6. PLOS authors have the option to publish the peer review history of their article (what does this mean?). If published, this will include your full peer review and any attached files.

Reviewer #1: No

---

## [Author Response · Author response to Decision Letter 0]

9 Sep 2022

We have attached the response as a seperate document

---

## [Decision Letter · Decision Letter 1]

20 Sep 2022

Improved And Optimized Drug Repurposing For The SARS-CoV-2 Pandemic

PONE-D-22-06101R1

Dear Dr. Issac,

We’re pleased to inform you that your manuscript has been judged scientifically suitable for publication and will be formally accepted for publication once it meets all outstanding technical requirements.

Kind regards,

Carlo Vittorio Cannistraci

Academic Editor

PLOS ONE

Additional Editor Comments (optional):

Reviewers' comments:

Reviewer's Responses to Questions

**Comments to the Author**

1. If the authors have adequately addressed your comments raised in a previous round of review and you feel that this manuscript is now acceptable for publication, you may indicate that here to bypass the “Comments to the Author” section, enter your conflict of interest statement in the “Confidential to Editor” section, and submit your "Accept" recommendation.

Reviewer #1: All comments have been addressed

2. Is the manuscript technically sound, and do the data support the conclusions?

Reviewer #1: Yes

3. Has the statistical analysis been performed appropriately and rigorously? 

Reviewer #1: N/A

4. Have the authors made all data underlying the findings in their manuscript fully available?

Reviewer #1: Yes

5. Is the manuscript presented in an intelligible fashion and written in standard English?

Reviewer #1: Yes

6. Review Comments to the Author

Reviewer #1: Authors made a great job in assessing all the comments. Code is now available and results are compared with respect to the state of the art. I suggest to accept the work in this form.

7. PLOS authors have the option to publish the peer review history of their article (what does this mean?). If published, this will include your full peer review and any attached files.

Reviewer #1: **Yes: **Pietro Hiram Guzzi

---

## [Editor Report · Acceptance letter]

25 Jan 2023

PONE-D-22-06101R1 

Improved And Optimized Drug Repurposing For The SARS-CoV-2 Pandemic 

Dear Dr. Issac:

I'm pleased to inform you that your manuscript has been deemed suitable for publication in PLOS ONE. Congratulations! Your manuscript is now with our production department. 

Kind regards, 

on behalf of

Dr. Carlo Vittorio Cannistraci 

Academic Editor

PLOS ONE